# The Relationship between Walking Speed and Step Length in Older Aged Patients

**DOI:** 10.3390/diseases7010017

**Published:** 2019-02-02

**Authors:** Yuji Morio, Kazuhiro P. Izawa, Yoshitsugu Omori, Hironobu Katata, Daisuke Ishiyama, Shingo Koyama, Yoshihisa Yamano

**Affiliations:** 1Department of Rehabilitation, Faculty of Medical Sciences, Shonan University of Medical Sciences, Yokohama 244-0806, Japan; yoshitsugu.omori@sums.ac.jp; 2Cardiovascular stroke Renal Project (CRP), Kobe 654-0142, Japan; izawapk@harbor.kobe-u.ac.jp or izawapk@ga2.so-net.ne.jp; 3Department of International Health, Graduate School of Health Science, Kobe University, Kobe 654-0142, Japan; 4Department of Rehabilitation, St. Marianna University School of Medicine Hospital, Kawasaki 216-8511, Japan; h-katata@marianna-u.ac.jp (H.K.); koyama-pt@marianna-u.ac.jp (S.K.); 5Department of Rehabilitation, St. Marianna University School of Medicine, Toyoko Hospital, Kawasaki, 211-0063, Japan; ishiyama@marianna-u.ac.jp; 6Department of Rare Diseases Research, Institute of Medical Science, St. Marianna University School of Medicine, Kawasaki 216-8511, Japan; yyamano@marianna-u.ac.jp

**Keywords:** maximum walking speed, step length, elderly patients, muscle force

## Abstract

Compared with elderly people who have not experienced falls, those who have were reported to have a shortened step length, large fluctuations in their pace, and a slow walking speed. The purpose of this study was to elucidate the step length required to maintain a walking speed of 1.0 m/s in patients aged 75 years or older. We measured the 10 m maximum walking speed in patients aged 75 years or older and divided them into the following two groups: Those who could walk 1.0 m/s or faster (fast group) and those who could not (slow group). Step length was determined from the number of steps taken during the 10 m-maximum walking speed test, and the step length-to-height ratio was calculated. Isometric knee extension muscle force (kgf), modified functional reach (cm), and one-leg standing time (s) were also measured. We included 261 patients (average age: 82.1 years, 50.6% men) in this study. The fast group included 119 participants, and the slow group included 142 participants. In a regression logistic analysis, knee extension muscle force (*p* = 0.03) and step length-to-height ratio (*p* < 0.01) were determined as factors significantly related to the fast group. As a result of ROC curve analysis, a step length-to-height ratio of 31.0% could discriminate between the two walking speed groups. The results suggest that the step length-to-height ratio required to maintain a walking speed of 1.0 m/s is 31.0% in patients aged 75 years or older.

## 1. Introduction

The characteristics of the walking pattern of the elderly include a low walking speed, short step length, slight increase of the two-leg support period, slight decrease of leg lifting in the swinging period, increase in the step width, decrease in the amount of arm swinging, and instability during changes of direction [1]. In comparison with elderly people who have not experienced falls, those who have experienced falls were reported to have a shortened step length, large fluctuations in their pace, and a slow walking speed [2]. Those whose walking speed is below 1.0 m/s are at high risk for leg injury, hospitalization, and death [3]. In terms of traffic signal programming for pedestrians in Japan, a speed of 1.0 m/s or faster is required to cross the street within the programmed period [4]. Therefore, a walking speed of 1.0 m/s or faster is one important ability that is necessary to take part in variable activities such as shopping, hobbies, and work.

Walking speed is related to age, height, lower limb muscular force, balance ability, and lower extremity joint disorders [5,6]. Among these variables, step length can be cited as one indicator of walking stability. Step length is reported to shorten in proportion to aging, resulting in a decrease in walking speed [6,7,8,9], or leg muscle strength [10]. However, the threshold of the step length required to maintain a walking speed of 1.0 m/s or faster is unknown. Knowledge of this data would be useful to clarify tasks for walking acquisition, setting of training goals, and maintaining practical mobility methods, and would be valuable information for the construction of physical therapy exercise programs.

We hypothesized that there would be a threshold of step length required to maintain a walking speed of 1.0 m/s or more in patients aged 75 years or older. Therefore, the purposes of this study were 1) to elucidate the relationship between walking speed and step length, and 2) to determine the cut-off step length value required to maintain a walking speed of 1.0 or more in patients aged 75 years or older.

## 2. Materials and Methods

### 2.1. Study Design

In this cross-sectional study, consecutive Japanese patients aged 75 years or older who were prescribed physical therapy at the St. Marianna School of Medicine Hospital Rehabilitation Department from November 2007 to October 2012 were selected as participants. Patients with cardiovascular disease, respiratory disease, gastrointestinal disease, metabolic disease, neoplastic disease, and urological disease were included, but those with hemiplegia, dementia, joint pain, cognitive impairment, and insufficient cardiovascular response were excluded.

### 2.2. Data Selection and Extraction

In this study, we measured walking speed, step length, lower extremity muscle force, and balance ability. We retrospectively collected demographic and clinical details, including underlying diseases, age, height, and body mass index (BMI) from the patients’ medical records.

Maximum walking speed was measured over a 10 m distance on a flat floor with a total pathway length of 12 m. We instructed the participants to walk the 12 m distance as fast as possible without falling. The time taken to walk the 10 m distance in the middle of the 12 m-long walking path was measured using a stopwatch. The measurement was conducted twice and the shorter time was used. Based on the obtained walking speed, we divided the participants into two groups: Those with a walking speed of 1.0 m/s or faster (fast group) and those who were not able to do so (slow group). The step length was calculated from the number of steps taken during the 10 m-maximum walking speed measurement, from which we calculated the step length-to-height ratio (%). Lower extremity muscle force was measured as the isometric knee extension muscle force. This measurement was performed according to the procedure previously reported using a handheld dynamometer manufactured by the Anima Corporation (μTasMT-1 or μTasF-1) [11]. The measurement was done twice on both sides and the largest value obtained for each leg was used. Then, the average between the left and right sides per body weight was calculated [kgf/kg]. We performed the Modified Functional Reach Test (M-FRT) [12] (cm) and measured one-leg standing (OLS) time in sec as the indices of balance ability. The M-FRT [12] is a modified version of the Functional Reach Test originally developed by Duncan et al. [13]. The M-FRT was measured using a telescoping rod. The participants were asked to assume a normal stance in front of a wall and hold the rod in one hand. Then, the participants extended their predominant arm horizontally (at approximately 90° in front of their body) and placed the tip of the rod against the wall. In the same way as the original FRT, the participants were asked to reach as far forward as they could without losing their balance and to press and shorten the rod against the wall. The examiner calculated the difference in the length (cm) of the rod between before and after reaching [14]. The measurement of OLS time was done with the precipitants’ eyes open, and they were instructed to remain standing on one leg for as long as possible. We used a stopwatch to measure the time during which the participants’ foot was not touching the floor. The OLS time was measured up to a maximum of 60 s. The participants who could not raise a leg at all were assigned a value of 0 s [15]. Measurements were conducted twice on each of the left and right legs and the maximum value obtained was used as the OLS time.

### 2.3. Ethics

The study was approved by the Institutional Committee on Human Research of St. Marianna University School of Medicine (approval No. 1967). The study was explained to all participants prior to their participation to the study and all participants signed an informed consent statement.

### 2.4. Statistical Analysis

Differences between the fast and slow groups were estimated using the Chi-squared test and unpaired *t*-test. To analyze the relationship between maximum walking speed and step length-to-height ratio, Pearson’s moment correlation coefficient was used. A logistic regression analysis was used to independently test the factors related to the maintenance of a walking speed of 1.0 m/s or more. Furthermore, a receiver operating characteristic (ROC) curve was used to detect the suitable step length-to-height ratio value required to maintain a walking speed of 1.0 m/s or more. The sensitivity, false-positive rate (1-specificity), and proper diagnosis rate were calculated to detect a suitable cutoff value for the step length-to-height ratio. A *p* value of <0.05 was considered to indicate statistical significance for all tests. All statistical analyses were performed using SPSS Statistics 21.0 (IBM SPSS Japan, Inc., Tokyo, Japan).

## 3. Results

### 3.1. Clinical Characteristics of the Patients

Of 1411 consecutive patients prescribed physical therapy in our department, 293 met the inclusion criteria. We further excluded 32 of these 293 patients because of insufficient data to evaluate clinical characteristics. Thus, 261 patients aged 75 years or older (50.6% men) who were prescribed physical therapy in our department (mean age 81.6 ± 5.4 years) were included in the analysis (Figure 1).

The participants’ diseases included cardiovascular disease (*n* = 164), gastrointestinal disease (*n* = 27), respiratory disease (*n* = 24), metabolic disease (*n* = 12), malignant neoplastic disease (*n* = 10), urological disease (*n* = 9), and others (*n* = 15). Among the 261 participants, 119 were assigned to the fast group and 142 were assigned to the slow group. Table 1 shows the details of the participants by group. Significant differences were found between the two groups in age, height, BMI, knee extension muscle force value, M-FRT results, OLS time, and step length-to-height ratio (Table 1).

### 3.2. Risk Scores for Patients Able to Walk 1.0 m/s or Faster in the Maximum Walking Speed Test

The results of the logistic analysis are shown in Table 2. Knee extension muscle force and step length-to-height ratio were significant factors independently related to the maintenance of a walking speed of 1.0 m/s or more. The odds ratio of the knee extension muscle force was 1.18 (increased by 0.03 kgf/kg), and that of the step length-to-height ratio was 2.91 (increased by 3%). The product correlation coefficient between maximum walking speed and step length-to-height ratio was 0.88, indicating a significant relationship (*p* < 0.01) (Figure 2).

Figure 3 shows the results of the ROC curve analysis showing the sensitivity and specificity. The independent variable is classifying the groups based on a walking speed of 1.0 m/s or more, and the dependent variable is step length-to-height ratio. The area under the curve was 0.961, and the step length-to-height ratio was the only factor that could significantly differentiate between the fast and slow groups (*p* < 0.01). The circled point on the ROC curve indicates the step length-to-height ratio of 31.0%, which corresponded to a sensitivity of 95.8% and false-positive degree of 14.1% for which the sum of sensitivity and specificity was the highest. The accuracy of the step length-to-height ratio of 31.0% as a cutoff value to discriminate between the two walking speed groups was 90.4%.

## 4. Discussion

To our knowledge, this is the first study to elucidate the step length required to maintain a walking speed of 1.0 m/s in elderly patients aged 75 years or older. As a result, knee extension muscle force and step length-to-height ratio were determined to be factors significantly related to a walking speed of ≥1.0 m/s. A step length-to-height ratio of 31.0% could discriminate the patients with a walking speed of ≥1.0 m/s.

A walking speed of 1.0 m/s is an index for the ability to walk across a road for pedestrians nationwide in Japan [4]. In the elderly, a shortened stride length appears prior to a decrease in walking speed [16]. Therefore, having information about a patient’s step length could be indicative of a restriction in walking. Relationships between walking speed and knee extension muscle force have been conventionally reported [5,6,17,18,19]. Rantanen et al. [18] reported a strong relationship between isokinetic knee extension force and walking speed in 75-year-old men and women. Similar results are obtained even in Japanese. Omori et al. [19] reported the isometric knee extension strength needed for a walking speed of 1.0 m/s or faster in 156 elderly patients. However, objective measurement of knee extension muscle strength requires specific equipment and a specific setting. In contrast, we could determine a patient’s step length using a simple measuring tape, indicating the convenience of measuring step length.

In this study, the step length-to-height ratio of 31.0% was chosen as the cutoff value with the highest rates of sensitivity, specificity, and accuracy. A step length-to-height ratio of 32% was reported to be more efficient than that of 40% and 48% in patients with Parkinson’s disease [20]. Thus, we believe that a step length-to-height ratio of 31.0% or more should be the target value to maintain comfortable walking in elderly people. The odds ratio of the step length-to-height ratio was 2.91 for a 3% unit of change. Since it can be calculated that, for example, a participant with a height of 150 cm could possibly cross the road three times faster if the step length-to-height ratio can be increased by 4.5 cm, this can become a really useful clinical index.

There are several limitations in this study. Step length is generally reported to be related to walking speed [8,9]. It was previously reported that when step length is adjusted by height, the influences of leg length and sex differences disappear [9]. For this reason, the present study used the step length-to-height ratio. We also used the maximum walking speed as the index of walking speed. In general, the maximum walking speed declines more steeply than the comfortable walking speed with aging. The present study assessed the variable of maximum walking speed in subjects aged 75 years or over only. However, maximum walking speed may be different in subjects under the age of 75 years, and in these younger subjects, it will also be necessary to consider the relationship with comfortable walking speed.

## 5. Conclusions

We conclude that a step length-to-height ratio of 31.0% is required to maintain a walking speed of 1.0 m/s in patients aged 75 years or older. Based on the present results, it is possible to suggest a strategy to increase step length and provide walking training in patients with a step length-to-height ratio of <31.0%. However, it will be necessary to confirm the training effect though longitudinal intervention studies. In addition, to analyze the data more precisely, we might need to collect step length data using force plates and motion analysis devices.

## Figures and Tables

**Figure 1 diseases-07-00017-f001:**
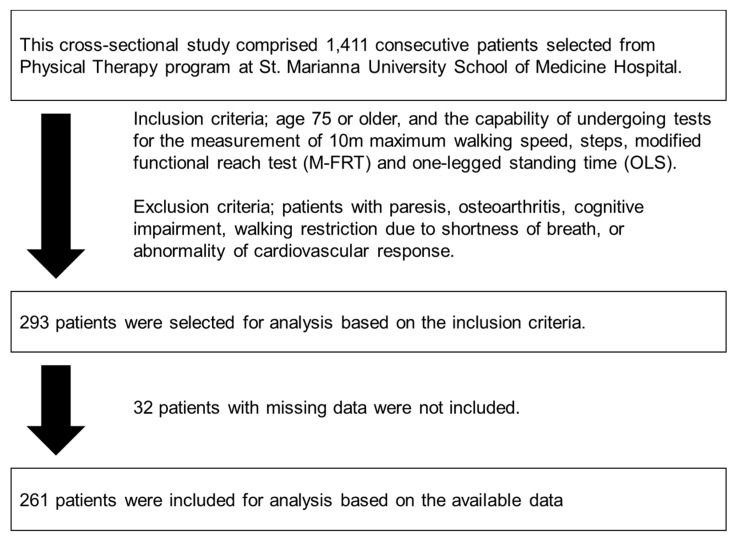
Diagram of the patient selection process.

**Figure 2 diseases-07-00017-f002:**
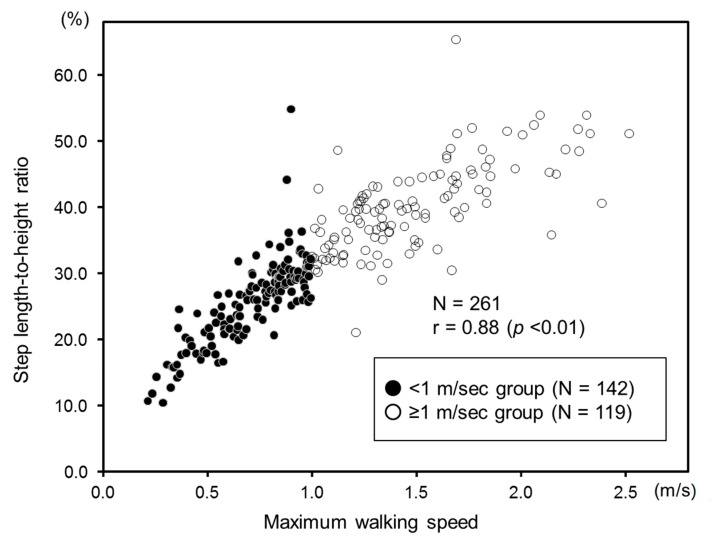
Relationship between the step length-to-height ratio and maximum walking speed. Step length-to-height ratio: Step length (cm)/height (cm).

**Figure 3 diseases-07-00017-f003:**
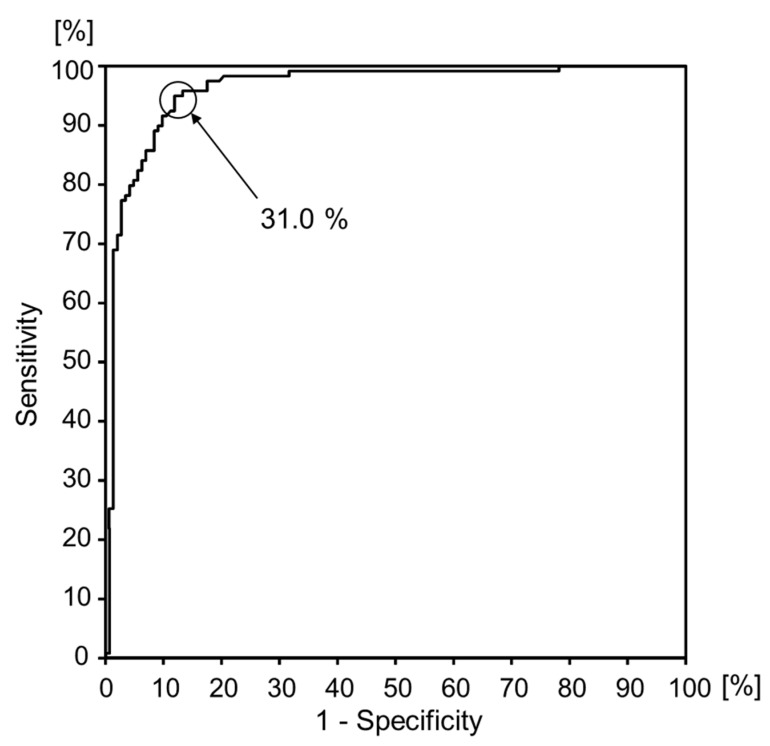
ROC analysis of walking speed in the ≥1.0 m/s group. The cutoff value for step length-to-height ratio identified by ROC analysis was determined to be 31.0%, with a sensitivity of 95.8%, 1-specificity of 85.9%, and area under the curve value of 0.96 (95% CI: 0.94–0.99, *p* < 0.01).

**Table 1 diseases-07-00017-t001:** Clinical characteristics of the patients.

Clinical Characteristics	≥1 m/s Group (N = 119)	<1 m/s Group (N = 142)	*p* Value
Age, years	80.2 ± 4.9	82.8 ± 5.5	<0.01
Male/Female, N	62/57	70/72	0.71 *
Height, cm	155.8 ± 8.7	153.4 ± 9.4	0.03
BMI, kg/m^2^	21.6 ± 2.9	19.9 ± 3.7	<0.01
Knee extension muscle force ^a^, kgf/kg	0.45 ± 0.15	0.29 ± 0.09	<0.01
M-FRT ^b^ (cm)	32.9 ± 5.7	26.0 ± 5.0	<0.01
OLS (s)	15.1 ± 18.0	2.0 ± 2.2	<0.01
Stride-to-height ratio (%)	40.0 ± 6.7	25.7 ± 6.3	<0.01

Values are mean ± standard deviation. ≥1 m/s: Group patients who could walk 1.0 m/s or faster in the 10 m walking speed test, <1 m/s: Group patients who could not walk 1.0 m/s or faster in the 10 m walking speed test, BMI: Body mass index, Knee extension muscle force: Average muscle force bilateral (kgf)/body weight (kg), OLS: One-leg standing time, Stride-to-height ratio: Step length (cm)/height (cm), ^a^ Knee extension muscle force: Average muscle force bilateral (kgf)/body weight (kg) (Katoh et al. 2009), ^b^ M-FRT: Modified Functional Reach Test (Morio et al. 2007). * χ^2^ value

**Table 2 diseases-07-00017-t002:** Risk scores for patients able to walk 1.0 m/s or faster in the maximum walking speed test.

Risk Scores for Patients	OR (95% CI)	*p* Value
Age (OR per 3 years)	1.21 (0.89–1.63)	0.23
Height (OR per 3 cm)	0.98 (0.85–1.14)	0.83
BMI (OR per 1 kg/m^2^)	1.04 (0.92–1.17)	0.51
Knee extension muscle force ^a^ (OR per 0.03 kgf/kg)	1.18 (1.01–1.37)	0.03
M-FRT ^b^ (OR per 3 cm)	1.10 (0.80–1.52)	0.56
OLS (OR per 3 s)	1.44 (0.95–2.18)	0.08
Stride-to-height ratio (OR per 3%)	2.91 (2.03–4.15)	<0.01

OR: Odds ratio, CI: Confidence interval, BMI: Body mass index, OLS: One-leg standing time, Stride-to-height ratio: Step length (cm)/height (cm), ^a^ Knee extension muscle force: Average muscle force bilateral (kgf)/body weight (kg) (Katoh et al. 2009), ^b^ M-FRT: Modified Functional Reach Test (Morio et al. 2007).

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
