# Peer review of "The Relationship between Walking Speed and Step Length in Older Aged Patients"

_diseases, 2019, doi:10.3390/diseases7010017_

Round 1

Reviewer 1 Report

A well written and conducted study.

Just a couple of editorial issues:

Line 119: SPSS should be cited (IBM SPSS Statistics for Windows, Version 21.0. Armonk, NY: IBM Corp.)

Figure 2: x-axis – “waling” needs to be “walking”

Author Response

Reply to Reviewer #1:

Thank you very much for your review and for your helpful comments. We have revised this manuscript on the basis of your and another reviewer’s comments. The revised paper has been edited for grammar and journal style by an expert in English medical writing prior to resubmission. My responses to your comments are as follows:

A well written and conducted study.

Just a couple of editorial issues:

Line 119: SPSS should be cited (IBM SPSS Statistics for Windows, Version 21.0. Armonk, NY: IBM Corp.)

Revise to

All statistical analyses were performed using SPSS Statistics 21.0 (IBM SPSS Statistics for Windows, Version 21.0. Armonk, NY: IBM Corp.).

Figure 2: x-axis – “waling” needs to be “walking”

Revise to

We have revised this to ‘Walking’

Reviewer 2 Report

Although the paper is adequately written, I am not sure what the clinical implications of these findings are for the patients. I think it is difficult to "impose" a step length on patients, especially in the elderly. Or do you think of performing specific kinesiology on these patients to improve their walking, and to decrease their risk of falling? Please elaborate

Author Response

Reply to Reviewer #2:

Thank you very much for your review and for your helpful comments. We have revised this manuscript on the basis of your and another reviewer’s comments. The revised paper has been edited for grammar and journal style by an expert in English medical writing prior to resubmission. My responses to your comments are as follows:

Although the paper is adequately written, I am not sure what the clinical implications of these findings are for the patients. I think it is difficult to "impose" a step length on patients, especially in the elderly. Or do you think of performing specific kinesiology on these patients to improve their walking, and to decrease their risk of falling? Please elaborate

Thank you for your suggestion. Yes, I do, I agree with you, this is very important information to know, we have already published about balance training and methods of evaluation of this previously in Japanese. So we are planning to do exercise training such as balance exercise and re-evaluate in a future study.

Koyama S, Morio Y, et al. The Relation of the Two-Square Step Test to Activities of Daily. Living and Walking Ability in Elderly Hospitalized Patients. Rigakuryoho gaku, 42(6), 480-486, 2015.

Reviewer 3 Report

The authors provide novel data regarding walking step length in the elderly. The work is well structured and presented. I believe that it would be a substantial contribution to the field.

Just some minor technical points

- Could the authors specify why patients undergoing physiotherapy were chosen?

- It would be useful to define what kind of physiotherapy, as well as the "ability" to complete the experimental procedure.

- Could the authors specify why relative and not absolute step length ratio was applied?

- Some information and advice for height recording, which can be tricky in the elderly.

- Could the authors specify why the fastest walk or the strongest muscle output were chosen for analysis, instead of the average? The same for the OLS time.

L49 related to

L107 participation to the study?

Regarding the discussion, it appears quite limited. Because of the literature in the field, as gait speed is usually presented, the authors could elaborate on such findings and comparisons.

Author Response

Reply to Reviewer #3:

Thank you very much for your review and for your helpful comments. We have revised this manuscript on the basis of your and another reviewer’s comments. The revised paper has been edited for grammar and journal style by an expert in English medical writing prior to resubmission. My responses to your comments are as follows:

The authors provide novel data regarding walking step length in the elderly. The work is well structured and presented. I believe that it would be a substantial contribution to the field.

Just some minor technical points

- Could the authors specify why patients undergoing physiotherapy were chosen?

Thank you for your suggestion. Yes, I agree with your great comments, so we need to find out and consider these in the future trial. At our facility, physiotherapy is mainly done at the request of a doctor.

- It would be useful to define what kind of physiotherapy, as well as the "ability" to complete the experimental procedure.

Thank you for your suggestion. Yes, I agree with your great comments, so we need to find out and consider these in the future trial.

- Could the authors specify why relative and not absolute step length ratio was applied?

Thank you for your suggestion. Yes, I agree with your great comments.There are differences in stride length of subjects, depending on height. Therefore, we utilized relative indicators.

- Some information and advice for height recording, which can be tricky in the elderly.

Thank you for your suggestion. Yes, I agree with your great comments, so we need to find out and consider these in the future trial.

- Could the authors specify why the fastest walk or the strongest muscle output were chosen for analysis, instead of the average? The same for the OLS time.

Thank you for your suggestion. Yes, previous many studies were chosen the fastest walk and/or the strongest muscle output in Japan. Because, there were maximum ability for patients, so we also used these values.

L49 related to

Yes, we have revised this to ‘related to”

L107 participation to the study?

Yes, we have revised this to ‘The study was explained to all participants prior to their participation to the study’

Regarding the discussion, it appears quite limited. Because of the literature in the field, as gait speed is usually presented, the authors could elaborate on such findings and comparisons.

Thank you for your suggestion. Yes, I do, I agree with you, this is very important information to know.

Round 2

Reviewer 2 Report

Thank you for the adapted information and the provided reference. I have no further comments.